

# Molecular characterization of the giant freshwater prawn (*Macrobrachium rosenbergii*) *beta-actin* gene promoter

Junjun Yan[1,2], Qiang Gao[3], Zongbin Cui[1], Guoliang Yang[4] and Yong Long[1]

[1] State Key Laboratory of Freshwater Ecology and Biotechnology, Institute of Hydrobiology, Chinese Academy of Sciences, Wuhan, China
[2] University of the Chinese Academy of Sciences, Beijing, China
[3] Zhejiang Institute of Freshwater Fisheries, Huzhou, China
[4] Huzhou University, Huzhou, China

## ABSTRACT

Constitutive promoters are important tools for gene function studies and transgenesis. The *Beta-actin* (*actb1*) gene promoter has been isolated from many species but remains to be cloned from the giant freshwater prawn (*Macrobrachium rosenbergii*). In this study, we cloned and characterized the *Mractb1* gene promoter. Two alternative promoters were identified for the *Mractb1* gene, which direct the generation of two transcripts with different 5′ untranslated regions. Three CpG islands were predicted in the upstream sequence, which are intimately related to transcription initiation and promoter activity. In addition to the CCAAT-box and the CArG-box, molecular dissection of the flanking sequence revealed the existence of one negative and two positive elements in the upstream region and the first intron. Finally, the *Mractb1* promoter demonstrated comparative activity to the carp (*Cyprinus carpio*) *actb1* promoter. Our investigations provide a valuable genetic tool for gene function studies and shed light on the regulation of the *Mractb1* gene.

## INTRODUCTION

Giant freshwater prawn is the biggest freshwater prawn in the world. The males can grow up to 320 mm in body length and weigh over 200 g (*Ling, 1969*). It has a wide distribution throughout the world and is particularly favored for farming in the tropical and subtropical developing countries in Asia, such as China, India and Thailand. Owing to its fast growth rate and higher profit realization per unit area, it has now become the most widely aquacultured freshwater prawn species and contributed 51.7% to the total global production of freshwater prawns (*Banu & Christianus, 2016*; *New, 2005*; *New & Nair, 2012*). However, the outbreak of diseases is a major limitation facing the aquaculture industry, and several diseases such as white spot disease and white tail disease have been reported to cause mass mortality for giant freshwater prawn farming worldwide (*Bonami & Widada, 2011*; *Saurabh & Sahoo, 2008*).

Corresponding authors
Guoliang Yang, ygl0572@163.com, 02462@zjhu.edu.cn
Yong Long, longyong@ihb.ac.cn

Transgenic technology has been considered as a powerful and effective way to improve economic traits of aquaculture species (*Bennett & Jennings, 2013*; *Rasmussen & Morrissey, 2007*; *Sin, 1997*), through which pathogen-resistant strains of prawn could be generated to reduce or even avoid the disease-related economic losses. So far, successful gene transfer in freshwater prawn and several marine shrimp species have been reported (*Arenal et al., 2000*; *Liu et al., 2001*; *Preston et al., 2000*). However, most of the researches used expression vectors based on heterologous promoters such as the CMV (cytomegalovirus) and SV40 (simian vacuolating virus 40) promoter, the hobo transposable element (HFL1) and the carp *beta-actin* gene promoter. Studies about fish transgenesis have shown that promoters of fish origin were more efficient in driving expression of the target genes than those with mammalian or viral origins (*Alam et al., 1996*; *Hanley et al., 1998*). Moreover, negative concerns about the use of viral promoters to express transgenes have led to the generation of "all-fish" transgene constructs (*Rasmussen & Morrissey, 2007*). In accordance with the "all-fish" concept, promoters originated from prawn species have strong potential for use in prawn transgenesis and gene function studies.

*Beta-actin* is a housekeeping gene with ubiquitous expression. The promoter of *beta-actin* gene has been isolated from and characterized in a wide variety of species such as human (*Gunning et al., 1987*), chicken (*Kosuga et al., 2000*), amphioxus (*Feng et al., 2014*), teleosts (*Barman et al., 2015*; *Hwang et al., 2003*; *Kong et al., 2014*; *Liu et al., 1991*; *Noh et al., 2003*) and shrimps (*Shi et al., 2015*; *Shi et al., 2016*). Because the *beta-actin* gene promoter confers a high level of constitutive transcriptional expression, it is widely used for transgenesis in both plants and animals (*Cho et al., 2011*; *Hong et al., 2016*; *Kosuga et al., 2000*). Although the cDNA of the *beta-actin* gene has been cloned from giant freshwater prawn (*Zhu et al., 2005*), its promoter remains to be characterized. Here, we report the isolation and characterization of the giant freshwater prawn *beta-actin* (*Mractb1*) gene promoter (Mbap). Our results provide insights into the regulation mechanisms of the *Mractb1* gene and indicate the potential usage of Mbap in gene function and transgenesis studies for giant freshwater prawn and closely related species.

## MATERIALS AND METHODS

### Animals
The animal protocol of this study was approved by the Institutional Animal Care and Use Committee of Institute of Hydrobiology (Approval ID: Y341131501). Adult giant freshwater prawns of both sexes (body weight 20–30 g) were acquired from Zhejiang Institute of Freshwater Fisheries, Huzhou, China. After transportation, the prawns were acclimated to laboratory conditions for 2 weeks in a circulating fresh water system at 28 °C.

### Total RNA and genomic DNA extraction
The prawns were immersed in eugenol solution (0.125 mL/L in phosphate buffered saline) until loss of consciousness and then placed into ice-slurry for a 2-Step euthanasia procedure (*Leary et al., 2013*). After that their bodies were dissected, and total RNA was extracted from the gill, liver, muscle and intestine of both sexes, and from testis and ovary, using TRIZOL (Invitrogen) according to the manufacturer's instructions. For genomic DNA

extraction, about 100 mg muscle tissue was put into a 1.5 mL Eppendorf tube with 0.5 mL lysis buffer (10 mM Tris-Cl, 1 mM EDTA, 0.1% SDS, 200 μg/mL proteinase K, pH 8.0). The samples were incubated in 58 °C water bath for 6 h with gentle vibration. After that, the lysate was subjected to a traditional phenol/chloroform extraction procedure. The concentration of RNA and DNA samples was measured by a NanoDrop 8000 from Thermo Scientific and the quality of RNA and DNA samples was assessed by agarose gel electrophoresis.

## Genome walking

Genome walking was performed as previously described (*Zhou et al., 2015*) to clone the 5′ flanking sequence of the *Mractb1* gene. Each walking step contains three rounds of nested PCR using different gene specific primers (GSP). Before genome walking, primers Mractb1-F and Mractb1-R (sequences of all the primers used in this study are listed in Table 1) designed according to the cDNA sequence deposited in GenBank (AY626840) were used to amplify partial genomic sequence. A 1,293-bp genomic DNA fragment was obtained and sequenced. The gene specific primers from Mractb1-GSP1 to Mractb1-GSP3 were subsequently designed according to this known genomic sequence. These gene specific primers were sequentially mated with the four degenerate primers included in the genome walking kit (Takara, Kusatsu, Japan) to amplify the unknown 5′ flanking sequence. Similarly, gene specific primers from Mractb1-GSP4 to Mractb1-GSP6 based on the product of the first walking experiment were used for the second walking step. A total of three walking experiments were conducted to identify the 5′ flanking sequence of the *Mractb1* gene.

## 5′ RACE

5′ RACE (rapid amplification of cDNA ends) was performed using the Takara 5′-Full RACE Kit according to the manufacturer's instructions. Briefly, the RNA sample was first treated with CIAP (calf intestine alkaline phosphatase) to remove the naked phosphorous from the RNA molecules, followed by TAP (tobacco acid pyrophosphatase) treatment for mRNA decapitation. The decapped mRNA was ligated to the 5′ RACE adaptor using T4 RNA ligase. Reverse transcription was performed using M-MLV (moloney murine leukemia virus) reverse transcriptase and 9-mer random primer. After that, two rounds of nest PCR using 5′-RACE- outer-primer/GSP1 and 5′-RACE-inner-primer/GSP2 primer pairs were conducted sequentially to clone the 5′ cDNA ends. Product of the second round PCR was purified using the Biospin Gel Extraction Kit (BioFlux), subcloned into the pMD18-T vector (TaKaRa) and then sequenced by the Tsingke Biological Technology (Wuhan, China) Co., Ltd.

## Quantitative real-time PCR

Quantitative real-time PCR (qPCR) was performed as previously described (*Long et al., 2013*) to determine the expression level of the *Mractb1* transcripts. Primers for qPCR were designed using the Primer Premier 6.0 software. First-strand cDNA was synthesized using total RNA extracted from different tissues by the RevertAid First Strand cDNA Synthesis

**Table 1** Sequences of primers.

| Primer name | Sequence (5′–3′) | Note |
|---|---|---|
| Mractb1-F | TTCCCATCCATTGTCGGCAG | cDNA amplification |
| Mractb1-R | GCATTCTGTCAGCGATTCCTGG | cDNA amplification |
| GSP1 | TGACCCATACCAACCATCACAC | 5′-RACE |
| GSP2 | ACCGGAGCCATTGTCTACAACCAAC | 5′-RACE |
| Exon1-F | ACTCGCTCTTCGACATC | cDNA cloning |
| Exon1′-F | AGTCTGTCACTTGCTCC | cDNA cloning |
| Exon3-R | TGTCTGTAGAAATGAATTTATTC | cDNA cloning |
| Mractb1-SP1 | GTACTGATATGAAGGCGTGTTCAG | Genome walking |
| Mractb1-SP2 | GATAATGATGGTAAGGCAAACATTG | Genome walking |
| Mractb1-SP3 | TACACCTGGAGTGTCTAAGCAG | Genome walking |
| Mractb1-SP4 | CTACTCCTGAAGATGTCGAAGAGCGAGTG | Genome walking |
| Mractb1-SP5 | TAACATCTGAAATGAAAGCGGACGAAACTG | Genome walking |
| Mractb1-SP6 | CAAACGTCTTGCCTTATATGGACATGGAG | Genome walking |
| Mractb1-SP7 | GTAGAAAGACCGGGATTTCTTTCGGT | Genome walking |
| Mractb1-SP8 | ACTGGGCGTAACTACTATGCCTCTAA | Genome walking |
| Mractb1-SP9 | GTTGAAGGGAAATGTACTGAGAACA | Genome walking |
| QMractb1-F1 | TCAGGAGTAGCACGTACAC | qPCR |
| QMractb1-F1′ | ATCACTGGTGCTCGTTG | qPCR |
| QMractb1-R0 | ACAATGGATGGGAACAC | qPCR |
| QMr18S-rRNA-F | TAGTTGGAGGTCAGTTCC | qPCR |
| QMr18S-rRNA-R | ATTCCAGAGTAGCCTGC | qPCR |
| Mractb1P1-F | ACT GAGCTCTCCCGAAGTGATCACTG | Promoter analysis |
| Mractb1P1-R | AGACT GCTAGCTTTGTATTAGCTGCAAGAGAAAG | Promoter analysis |
| Mractb1P2-F | ACT GAGCTCTCATTTAGTAAGTAGGAGAG | Promoter analysis |
| Mractb1P2-R | AGACT GCTAGCAGCAAGTGACAGACTGAAC | Promoter analysis |
| Mractb1P3-F | ACT GAGCTCAGGTAAGTACACGTTGGC | Promoter analysis |
| Mractb1P3-R | AGACT GCTAGCTGAGATAAATTATGGAAC | Promoter analysis |
| Mractb1P4-F | ACT GAGCTCTTTGCCCTTCCGCGAAATTTAC | Promoter analysis |
| Mractb1P4-R | AGACT GCTAGCGGTGAGAGTGTACGTGCTAC | Promoter analysis |
| Mractb1P5-F | ACT GAGCTCACATTATGGAAACATTTC | Promoter analysis |
| Mractb1P5-R | AGACT GCTAGCATTCTTTGCATGTGACGAG | Promoter analysis |
| Mractb1P6-F | ACT GAGCTCATGATGAGGGTCACGCGTTAAG | Promoter analysis |
| Mractb1P6-R | AGACT GCTAGCTTCGGCGAACTGGGCGT | Promoter analysis |
| Mractb1P7-F | ACT GAGCTCTGCGCTGCTTTTACCAAATAC | Promoter analysis |
| Mractb1P7-R | AGACT GCTAGCTCATCATCTTTACCATC | Promoter analysis |
| Mractb1P42-F | TACACTCTCACCTCATTTAGTAAG | Promoter analysis |
| Mractb1P24-R | CTTACTAAATGAGGTGAGAGTGTA | Promoter analysis |
| Carpactb1-F1 | ACG AAGCTTTCAAACTGTGGCACCATC | Promoter analysis |
| Carpactb1-R1 | ACG CCATGGCTGAACTGTAAATGAATG | Promoter analysis |

Kit (Thermo scientific) and used as template for qPCR analysis. *Mr18S-rRNA* was used as the internal reference for gene expression normalization.

## Bioinformatic analyses

CpG islands in the upstream region were predicted by the Methprimer software using default parameters (http://www.urogene.org). Core promoter elements were predicted by the YAPP Eukaryotic Core Promoter Predictor (http://www.bioinformatics.org/yapp/cgi-bin/yapp.cgi).

## Construction of plasmids

To dissect transcriptional activity of the 5′ flanking sequence of the *Mractb1* gene, the cloned upstream sequences were divided into seven fragments according to location of the predicted GC islands and the TSSs (transcriptional start sites) revealed by 5′ RACE. The position of the first nucleotide of the initiation codon was regarded as +1. The length and relative location of fragments 1 to 7 are 558 bp (−558 to −1), 523 bp (−1,080 to −558), 240 bp (−1,317 to −1,078), 249 bp (−1,568 to −1,320), 195 bp (−1,763 to −1,569), 357 bp (−2,120 to −1,764) and 501 bp (−2,620 to −2,120), respectively. These fragments were amplified and cloned into the SacI/NheI site of the pGL3- Basic vector (Promega, Madison, WI, USA) separately or in combination. Deletion of fragment 3 in the 5′ flanking sequences was conducted via the PCR-driven overlap extension method as previously described (*Heckman & Pease, 2007*) with primers Mractb1P42-F and Mractb1P24-R. The constructs were named according to the inserted fragments, for example, pGL-Mba7654321 includes the whole sequence, pGL-Mba1 contains only fragment 1 and pGL-Mba21 consists of both fragment 1 and fragment 2. To compare the promoter activity of *Mractb1* gene with that of carp *actb1* gene, the promoter of carp *actb1* gene was amplified by the primer pair Carpactb1-F1/Carpactb1-R1 and inserted into the HindIII/NcoI site of the pGL3-Basic vector.

## Luciferase activity assay

The EPC (endothelial progenitor cells) cells (ATCC® CRL-2872™) were maintained at 28 °C with 5% $CO_2$ in M199 medium (Hyclone, Logan, UT, USA) supplemented with 10% fetal bovine serum, 100 μg/mL streptomycin, 100 U/mL penicillin and 2 μg/mL amphotericin B. One day before transfection, cells were seeded into 24-well plates at a density of $2 \times 10^5$ cells/well. Transfection was performed using the X-tremeGENE HP Transfection Reagent (Roche). The pRL-TK plasmid from Promega was used as internal control for luciferase activity assays and co-transfected with the promoter-luciferase constructs. The total amount of plasmid used for transfection was 500 ng/well and the ratio between the promoter-luciferase vector and pRL-TK was 5:1. Twenty-four hours after transfection, cells were lysed and subjected to luciferase activity assays using the Dual-Luciferase® Reporter Assay System from Promega.

## Western blot

EPC cells cultured in six-well plates were transfected with 2 μg plasmid. Twenty-four hours after transfection, cells were lysed by the RIPA (radio immunoprecipitation assay) lysis

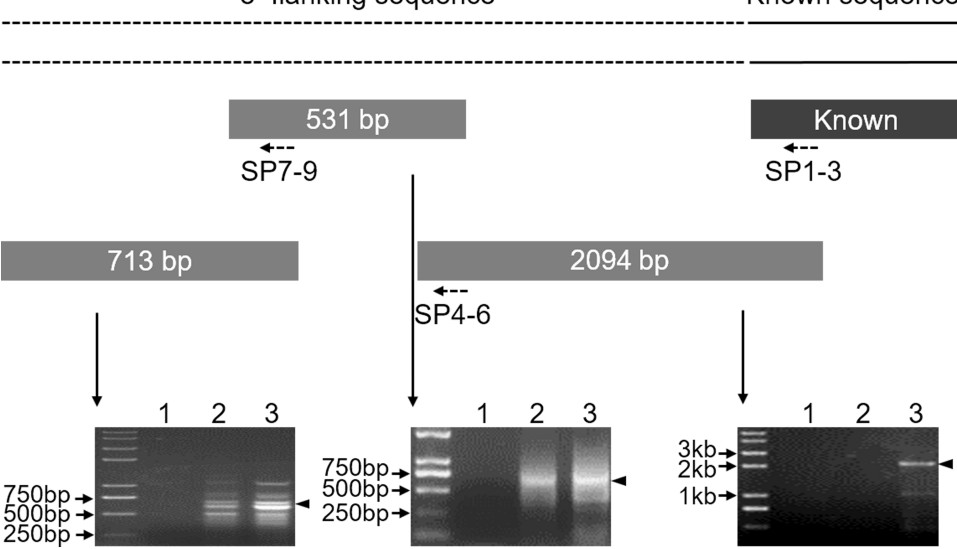

**Figure 1** **Cloning of the 5′-flanking sequence of the *Mractb1* gene.** The 5′-flanking sequence of *Mractb1* was cloned by three rounds of genome walking based on asymmetric interlaced PCR. The dashed line represents the assembled 5′ flanking sequence. The rectangular blocks and the dotted arrows indicate the cloned sequences and the primers used for each round of genome walking respectively. The electrophoretograms were displayed accordingly. The numbers above the electrophoretograms indicate different round of PCR for each walking step.

buffer from beyotime. Then western blot was performed as previously described (*Mo et al., 2010*) and GAPDH (glyceraldehyde-3-phosphate dehydrogenase) was used as loading control. The primary antibodies used for western blot were rabbit anti-Firefly Luciferase (Abcam, #ab185923, 1:5000) and mouse anti-GAPDH (Boster, #BM3876, 1:100). The secondary antibodies were goat anti- rabbit IgG (immunoglobulin) (Boster, #BM3894, 1:5000) and goat anti-mouse IgG (#BM3895, 1:5000; Bosterbio, Pleasanton, CA, USA), respectively. Intensity of the bands in the western blot image was analyzed by the ImageJ software (http://rsb.info.nih.gov/ij/).

## Statistical analysis

The results of dual luciferase assays and qPCR are presented as mean ± standard deviation. Independent samples $t$-tests were performed using SPSS 15.0 (Chicago, IL, USA) to analyze the significant difference between groups.

## RESULTS

### Cloning of the 5′ flanking sequence for the *Mractb1* gene

To clone the 5′ flanking sequence for the *Mractb1* gene, a 1,293 bp genome sequence was firstly amplified using the primers Mractb1-F/ Mractb1-R designed according to the reported cDNA sequence (*Zhu et al., 2005*). Three rounds of genome walking experiments were subsequently performed to clone the upstream sequence of the *Mractb1* gene. As shown in (Fig. 1), the length of the DNA fragments obtained by the three genome walking

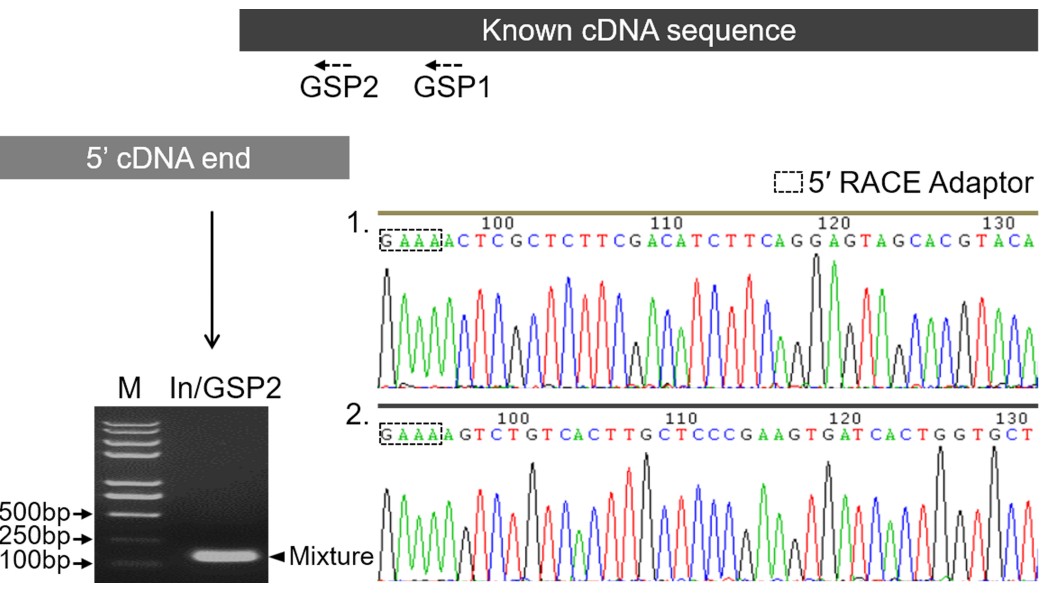

**Figure 2  Determination of the transcription start sites of the *Mractb1* gene.** The shaded rectangles represent the known cDNA sequence (black) and the 5′ cDNA end sequence obtained by 5′ RACE (grey), respectively. The electrophoretogram shows the 5′ RACE product (indicated by an arrowhead). The base peak maps indicate the two 5′ cDNA end sequences identified by DNA sequencing. M, DNA marker; GSP1 and GSP2, gene specific primers; In, 5′ RACE inner primer.

experiments were 2,094, 531 and 713 bp, respectively. Furthermore, PCR using the forward primer Mractb1-F and the reverse primer Exon3-R (matches to the 3′ cDNA end) generated a 1,593 bp sequence. Assembling of these fragments resulted in a 4,306-bp long contig, which contains both the 5′ flanking and full sequence for the *Mractb1* gene. The obtained upstream sequence counted from the first nucleotide of the initiation codon is 2,620 bp in length, which has been submitted to the GenBank database under the accession number KY038927.

## Identification of two transcripts for the *Mractb1* gene

Before characterizing the promoter of the *Mractb1* gene, 5′-RACE was performed to identify the transcriptional start site (TSS). A clear DNA band in the range of 100 to 250 bp was obtained (Fig. 2). The DNA fragments were purified and cloned into the pMD18-T vector. Subsequent DNA sequencing revealed two 5′ cDNA end sequences for the *Mractb1* gene. One is 106 bp (10 clones sequenced) and the other is 145 bp (6 clones sequenced) in length (Fig. 2). Further analysis revealed that the *Mractb1* gene possesses 2 transcript variants (designated Mractb1_tv1 and Mractb1_tv2, respectively), which only differ in the 5′ UTR (untranslated region sequence) (Fig. 3A). PCR using different forward primers (Exon1-F and Exon1′-F) and the same reverse primer (Exon3-R) amplified the full-length sequence for the 2 transcripts (Figs. 3A, 3B), which are 1,186 and 1,225 bp in length, respectively. Since the transcripts contain different exon 1 (Fig. 3A), we speculate that they are driven by alternative promoters (promoter-1 and promoter-2).

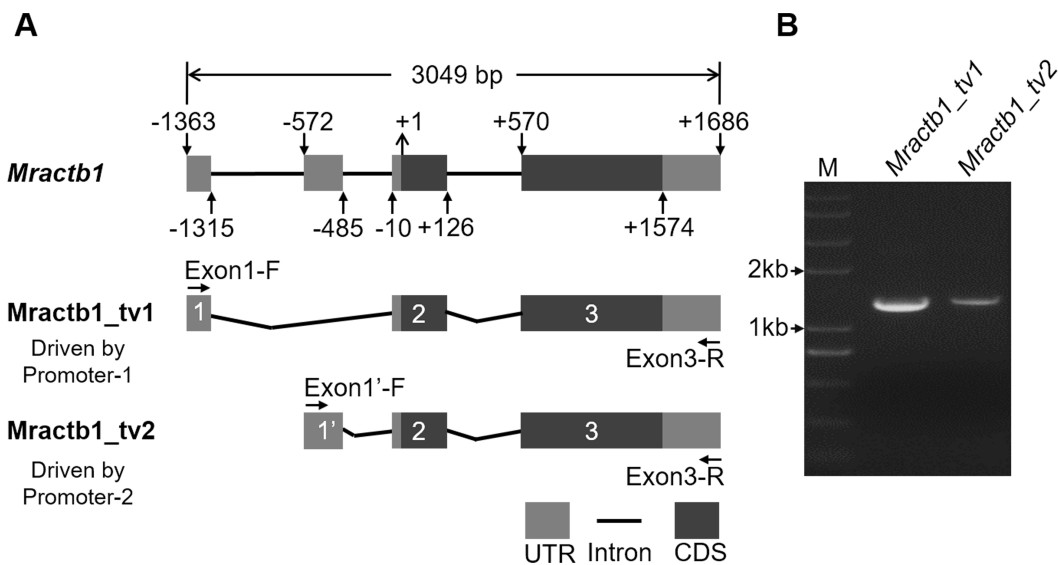

**Figure 3  Transcripts of the *Mractb1* gene.** (A) Structure of the *Mractb1* gene and its two transcript variants with different initial exon. The numbers and arrows indicate the relative position of the exons and coding sequence (CDS). The first base of the start codon was defined as position +1. The numbers in the rectangles indicate different exons of the Mractb1 gene. The alternative first exons are designated as exon 1 and exon 1′. (B) Electrophoretogram for the two transcripts. Primers used to amplify the two transcripts are Exon1-F, Exon1′-F and Exon3-R as displayed in (A). M: DNA marker.

## Expression of the *Mractb1* transcripts in different sexes and tissues

To shed light on the activities of the promoters in different sexes and tissues, the expression levels of the two transcript variants were measured by qPCR assays. As shown in Fig. 4, the *Mractb1* gene demonstrate quite different expression levels among tissues and between sexes. The highest overall expression level of the *Mractb1* gene was found in the gonad and gill, followed by the intestine, muscle and liver. Sexual dimorphism could be found for the expression level of the *Mractb1* gene in all the tissues (Fig. 4). The expression levels of both Mractb1_tv1 and Mractb1_tv2 in the female gonad (ovary) and gill is significantly higher than those in the corresponding male tissues, while muscle and intestine of the males demonstrated higher expression than those of the females (Fig. 4). A difference between the expression of the two transcripts in the same tissue was identified in the male gill and the female gonad and muscle. Mractb1_tv2 showed a higher expression in the male gill, but a lower expression in the female muscle and gonad when compared to Mractb1_tv1 (Fig. 4). These findings indicate that expression of the *Mractb1* gene is regulated by tissue and sex specific factors.

## Molecular dissection of the *Mractb1* gene promoter

Three CpG islands were predicted in the 5′ flanking sequence of the *Mractb1* gene. These CpG islands are located at −1,762 to −1,569 (i1, 194 bp), −1,437 to −1,320 (i2, 118 bp) and −747 to −542 (i3, 206 bp), respectively (Fig. 5A). The 2,620 bp flanking sequence was divided into 7 segments (from Mba7 to Mba1) according to the locations of the CpG

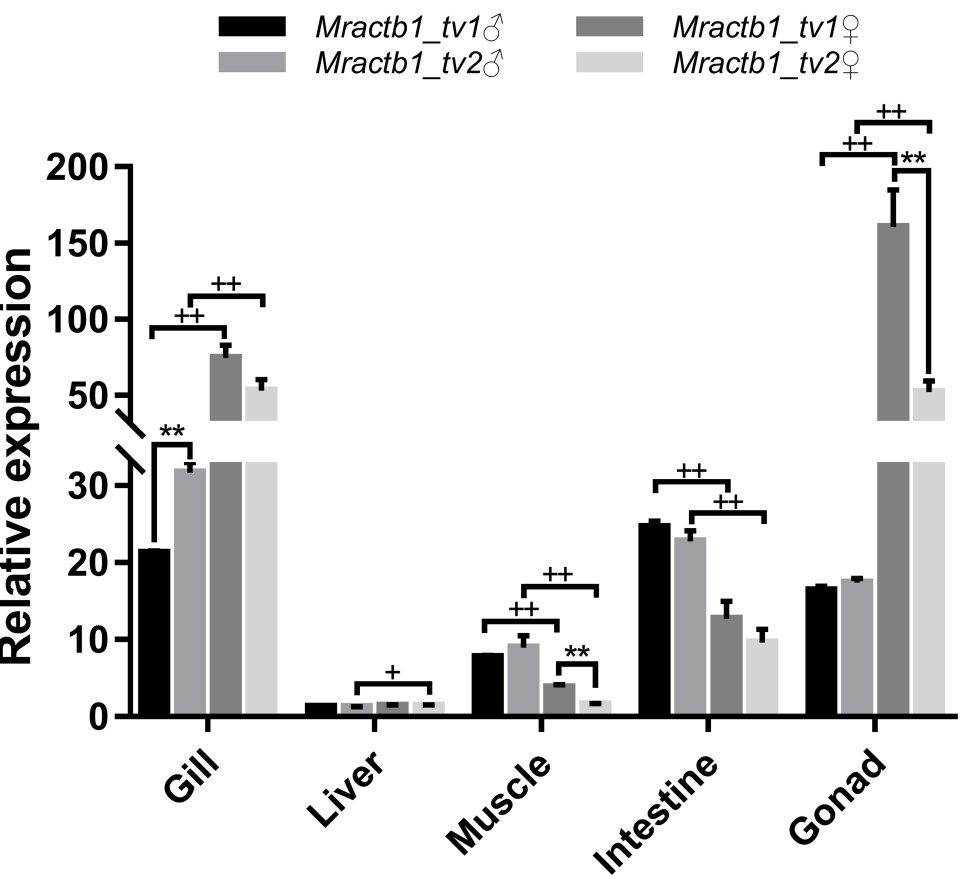

**Figure 4** Expression of the two *Mractb1* transcripts in different sexes and tissues.

islands and the TSSs (Fig. 5A). Transcriptional activities of the upstream fragments were investigated by dual luciferase assays. When characterized separately, Mba7, Mba6, Mba3 and Mba1 demonstrated no or minimal transcriptional activity, while high promoter activity was detected for Mba5, Mba4 and Mba2, each of them contains a predicted CpG island (Fig. 5B), suggesting importance of the CpG islands in transcription. Based on structure of the *Mractb1* transcripts and transcriptional activities of the upstream fragments, we concluded that the two alternative promoters are in Mba54 (promoter-1) and Mba21 (promoter-2), respectively. Activity of the two promoters and regulations by their corresponding upstream regions were further characterized. For promoter-1, combination of Mba5 and Mba4 only slightly increased the promoter activity, Mba6 displayed no effect on Mba54, while fusing Mba7 to Mba654 significantly enhanced the promoter activity (Fig. 5C). As for promoter-2, significant synergistic effect was found between Mba2 and Mba1; however, Mba3 demonstrated negative effect on the activity of Mba21 (Fig. 5D). Serial deletion experiments indicated that Mba654321 had the highest transcriptional activity, followed by Mba7654321 and Mba54321, while Mba4321, Mba321 and Mba21 only displayed basal promoter activity (Fig. 5E). Furthermore, interaction between the two alternative promoters was observed. Mba54 (promoter-1) and Mba21 (promoter-2)

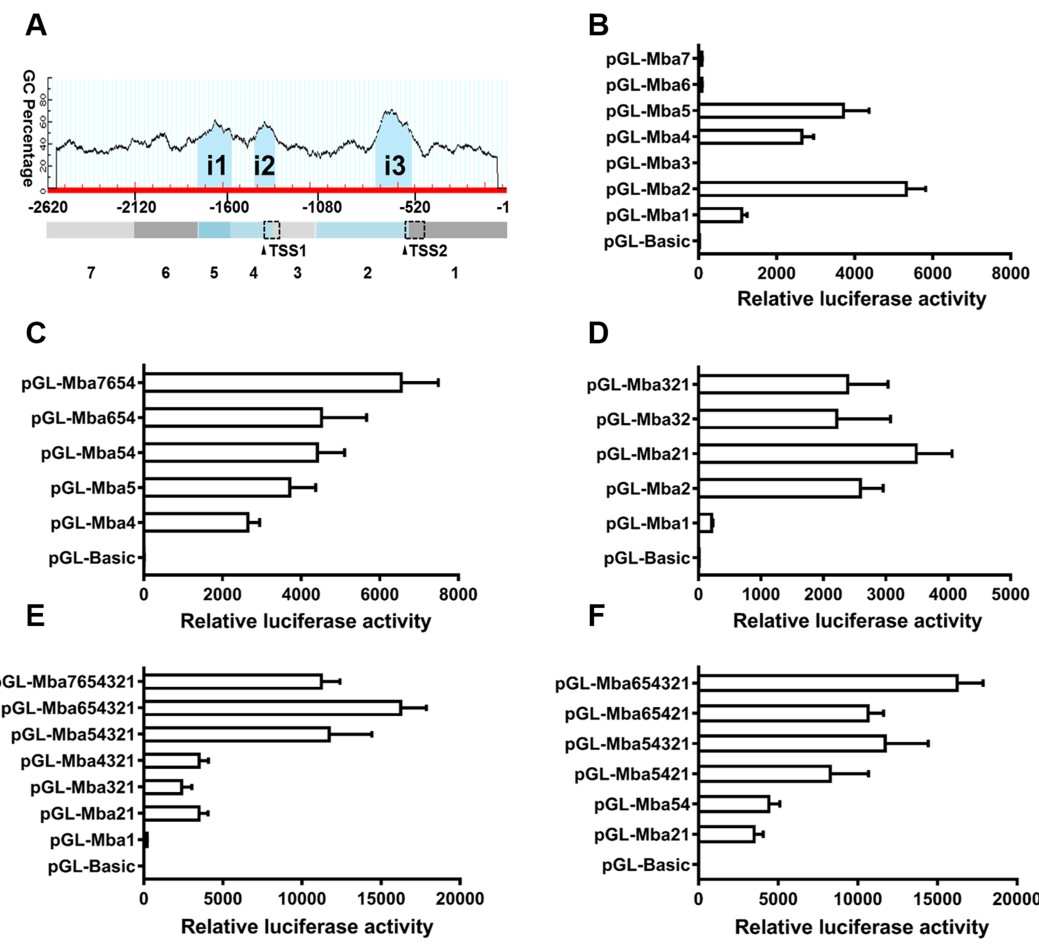

**Figure 5   Molecular dissection of the *Mractb1* gene promoter.** (A) Schematic diagram of the DNA fragments subjected to promoter activity assays. The 2,620 bp 5′-flanking sequence was divided into 7 fragments (rectangles shaded with different color) according to the location of the TSSs and CpG islands (i1: −1,762/−1,569, i2: −1,437/−1,320, i3: −747/−542) predicted by the Methprimer software (http://www.urogene.org/). The arrowheads indicate TSSs. The dashed rectangles represent first exons of the two transcripts. (B) Promoter activity of the flanking regions characterized individually. (C) Characterization of promoter-1. (D) Characterization of promoter-2. (E) Deletion analysis of the whole sequence. (F) Interaction between the two alternative promoters. The DNA fragments were cloned into the pGL3-basic vector to generate promoter activity analyzing plasmids. The constructs were transfected into EPC cells and the transcriptional activities were analyzed by dual luciferase assays. The relative luciferase activity represents the ratio between the firefly luciferase activity and the corresponding renilla luciferase activity. The bars indicate mean ±SD ($n = 3$).

demonstrated significant synergistic effects when combined (Mba5421). Both Mba3 and Mba6 can enhance activity of Mba5421 and the interaction between Mba3 and Mba6 is necessary for the full activity of the *Mractb1* gene promoter (Fig. 5F). Taken together, the upstream fragments demonstrate different effects under different circumstances. For example, Mba7 enhances activity of the promoter-1 but inhibits activity of the whole promoter (Figs. 5C, 5E). On the contrary, Mba3 inhibits activity of the promoter-2 but is indispensable for the full activity of the whole promoter (Figs. 5D, 5F). These findings

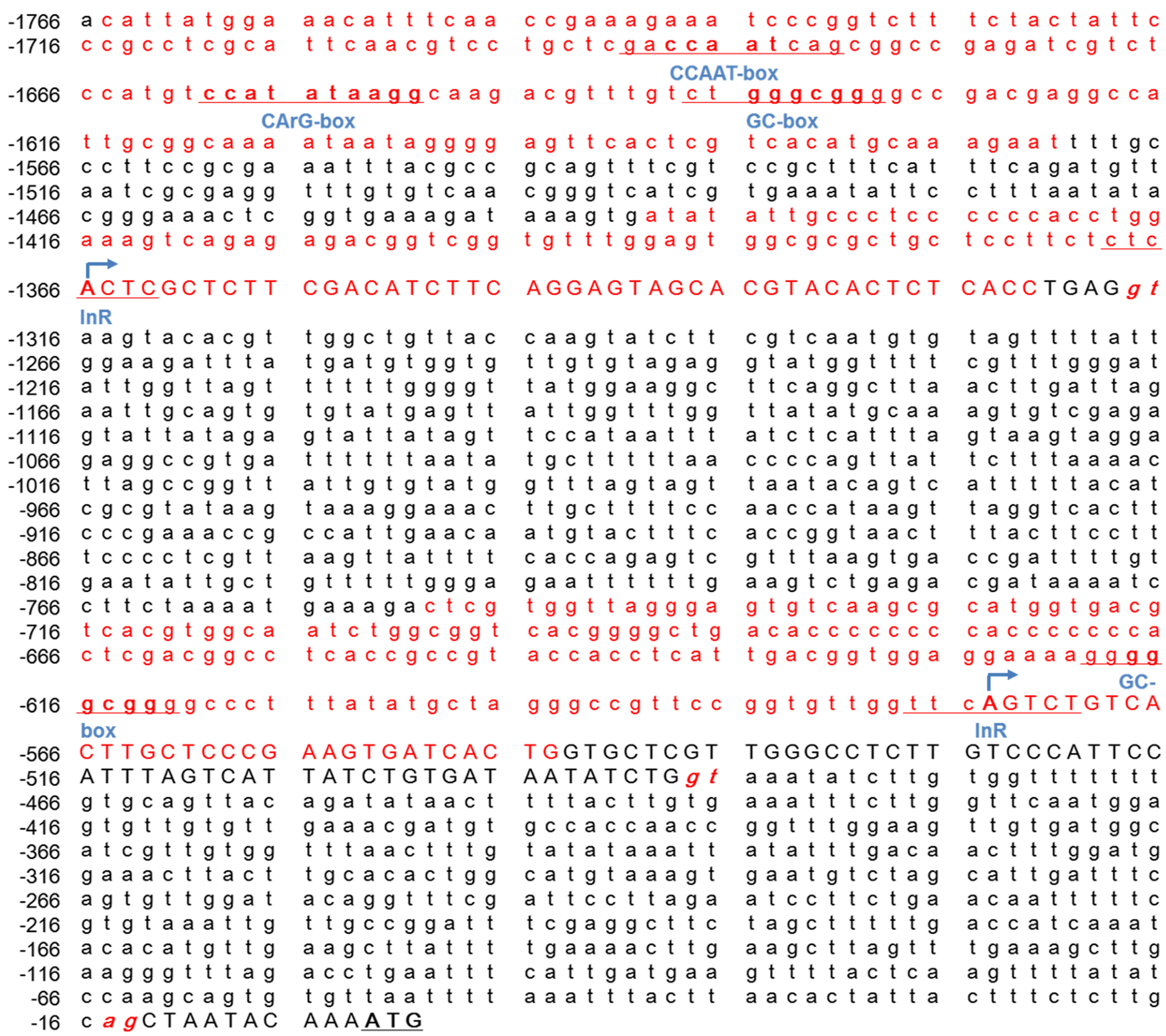

**Figure 6** **The conserved genetic elements in the *Mractb1* 5′-flanking region.** The conserved genetic elements in the *Mractb1* 5′-flanking region. The CpG islands are shown in red. The exons are shown in uppercase and the introns are shown in lowercase. The numbers represent relative position of the nucleotides, and the first base of the start codon is regarded as +1. The donor/acceptor sites for splicing are shown in red italics. The arrows indicate transcription start sites. The CCAAT-box, CArG-box, GC-box and InR are underlined.

suggest complex interactions among genetic elements in regulating expression of the *Mractb1* gene.

## Genetic elements in the *Mractb1* gene promoter

As shown in (Figs. 5A 5B), the CpG islands in the promoter of the *Mractb1* gene are intimately associated with the promoter activity. It is interesting that despite no TSS was identified in the first CpG island (i1, Mba5), it still directs high level of transcription (Fig. 5B). Further investigation indicated that this region contains a CCAAT box, a CArG box and a GC box (Fig. 6). Both CCAAT box and CArG box were reported to be important for the constitutive expression of carp and human *beta-actin* gene (*Liu et al., 1990*; *Quitschke et al., 1989*). The second CpG island (i2) encompasses the first TSS and an initiator element (InR) (Fig. 6). The third CpG island (i3) contains the second TSS, a GC box and an InR

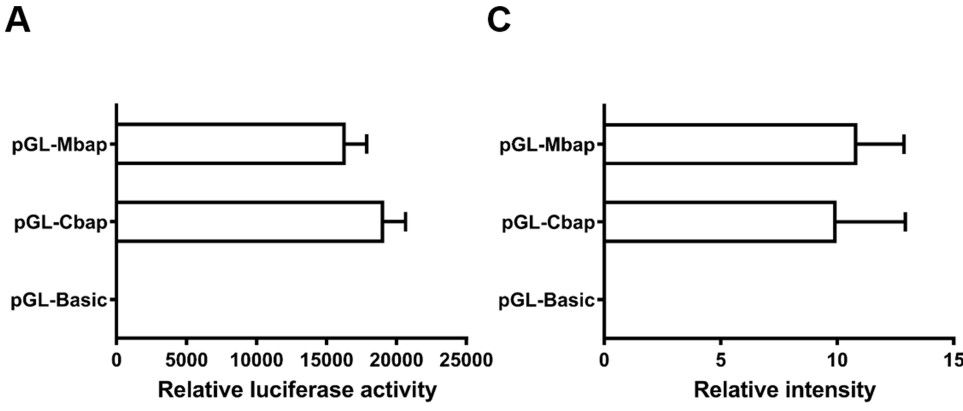

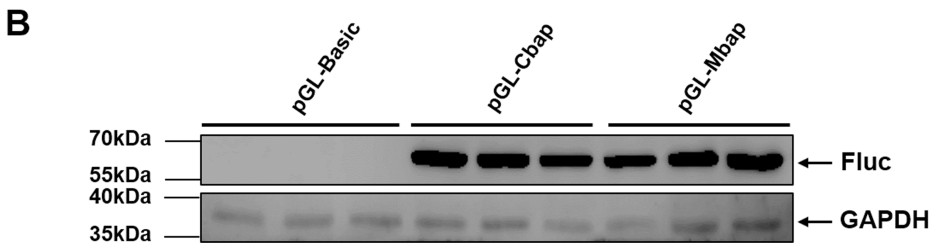

**Figure 7** *Mractb1* **promoter demonstrates comparative activity with carp** *actb1* **gene promoter.**
(A) Relative luciferase activity. (B) Western blot of firefly luciferase. (C) Relative levels of the firefly luciferase protein. The *Mractb1* promoter (Mbap) and carp *actb1* promoter (Cbap) were cloned into the pGL3-basic vector and the resulted constructs were transfected into EPC cells. Dual luciferase assays and western blot were performed to analyze the activities and protein levels of the luciferase reporter, respectively. GAPDH was used as loading control for western blot. The relative intensity represents ratio of the brightness of the firefly luciferase bands to those of GAPDH. The bars indicate mean $\pm$ SD ($n = 3$).

(Fig. 6). Since no TATA box was identified in the core promoters for the *Mractb1* gene, the InRs may play important roles in transcription initiation.

### *Mractb1* promoter demonstrates comparative activity with the carp *actb1* promoter

To justify whether the *Mractb1* promoter cloned in this study has the potential to be used as a genetic tool in gene function and transgenesis studies, the activity of which was compared with that of carp *beta-actin* (*actb1*) gene promoter, a promoter which is commonly used in transgenic fish. The fragment with the highest promoter activity (Mba654321) was designated as Mbap. The promoter of the carp *actb1* gene (designated as Cbap) was cloned into the pGL3-basic vector as well. Dual luciferase assays in EPC cells indicated that the Mbap demonstrated comparative activity with the Cbap (Fig. 7A). Western blot using antibody against the firefly luciferase detected similar expression level of firefly luciferase in the EPC cells transfected with pGL-Mbap and pGL-Cbap (Figs. 7B, 7C); however, the empty vector pGL-basic demonstrated no luciferase expression at all. These results indicate

that the *Mractb1* promoter is as active as the carp *actb1* promoter and which can be used for transgenesis and gene function studies for prawn and related species in the future.

## DISCUSSION

As a house-keeping gene, the promoter of *actb1* is cloned from a wide range of organisms and extensively used for gene function and transgenesis studies (*Barman et al., 2015*; *Cho et al., 2011*; *Hwang et al., 2003*; *Kong et al., 2014*; *Liu et al., 1991*; *Noh et al., 2003*). Although the CDS (coding sequence) and 3′ UTR sequence of the *Mractb1* gene have been cloned previously (*Zhu et al., 2005*), the promoter remains to be characterized. In this study, we cloned the upstream sequence of the *Mractb1* gene, identified the TSSs, investigated expression of the transcripts in different tissues of both sexes and characterized transcriptional activity of the *Mractb1* gene promoter. Both luciferase assays and western blot indicated that the *Mractb1* promoter demonstrated comparable transcriptional activity to that of the carp *actb1* promoter in the EPC cells, suggesting that it is a valuable tool for gene function and transgenesis studies for *Macrobrachium rosenbergii*.

Identification of two TSSs and the results of promoter activity assays indicated that the *Mractb1* gene possesses two alternative promoters which initiate transcription from different TSSs and lead to the generation of two transcript variants. These two transcript isoforms contain different initial exons but share the same open reading frame (ORF) and 3′ UTR, indicating that they only differ in the 5′ UTR. Existence of alternative promoters is common for mammalian genes. It was reported that 18% of human genes have evidence of alternative promoter usage (*Landry, Mager & Wilhelm, 2003*). Alternative promoters may differ in tissue and developmental stage specificity and transcriptional activity (*Landry, Mager & Wilhelm, 2003*). The usage of alternative promoters may account for distinct mRNA levels of the *Mractb1* gene transcripts in different tissues and sexes, and different 5′ UTR of the two transcript variants may affect their translation efficiency. Moreover, molecular dissection for the activities of different upstream regions revealed significant synergistic effects between the two alternative promoters. Although the *actb1* gene promoters of human and carp were well characterized, it is unknown whether they possess alternative promoters as well.

Three CpG islands were identified in the 5′ flanking sequence of the *Mractb1* gene. These CpG islands are closely related to the activity of the *Mractb1* gene promoter according to luciferase assays. All of them demonstrated transcriptional activity and both of the identified TSSs are located in the two proximal CpG islands, consistent to the discovery that most CpG islands function as sites of transcription initiation by destabilizing nucleosomes and attracting proteins that create a transcriptionally permissive chromatin state (*Deaton & Bird, 2011*). CpG islands are prevalent in the promoter of housing keeping (HK) genes and more than three quarters (78.7%) of human HK (housing keeping) genes predominantly have a CpG+/TATA- core promoter (*Zhu et al., 2008*). CpG-islands usually contain multiple GC-boxes, and GC-boxes coupled with InR elements are essential to initiate transcription in the absence of TATA-box (*Butler & Kadonaga, 2002*). Consistent with this notion, both GC-box and InR, but no TATA-box were found in the core promoters of the *Mractb1* gene.

Genetic elements including the CCAAT-box and CArG-box were reported to be important for the constitutive expression of human and carp *actb1* gene (*Liu et al., 1990*; *Quitschke et al., 1989*). These elements were also identified in the *Mractb1* promoter. Fusing the DNA fragment containing these elements (Mba5) to the proximal promoter sequence (Mba4321) dramatically increased the promoter activity (Figs. 5A, 5E), suggesting the function of these elements in regulating the *Mractb1* gene. As in the carp *actb1* gene promoter, a distal upstream region (Mba7) was found to negatively regulates the *Mractb1* promoter activity. Moreover, the upstream fragment Mba6 and 5′ end of the first intron Mba3, and the interaction between these two regions are necessary for the constitutive transcriptional activity of the *Mractb1* gene promoter. These results indicate that in addition to the conservative elements, other unknown elements and factors are involved in regulating the *Mractb1* gene.

## CONCLUSIONS

The *beta-actin* (*Mractb1*) gene promoter of *M. rosenbergii* was cloned and characterized. Two alternative promoters were identified for the *Mractb1* gene, which direct the generation of two transcripts with different 5′ UTR. Genetic dissection of the upstream sequence of the *Mractb1* gene revealed one distal negative element and two proximal positive elements regulating the activity of the *Mractb1* gene promoter. Finally, the *Mractb1* promoter demonstrated comparative activity to the carp (Cyprinus carpio) *actb1* promoter. Our investigations provide a valuable genetic tool for gene function studies and shed light on the regulation of the *Mractb1* gene.

## ACKNOWLEDGEMENTS

The authors would like to thank Dr. Bolan Zhou from the Hunan University of Arts and Science for technical assistance.

### Funding

This work was supported by the Youth Innovation Promotion Association of Chinese Academy of Sciences (No.2016305) and the Major Project for New Agricultural Varieties Breeding of Zhejiang Province (No.2012C12907-2). The funders had no role in study design, data collection and analysis, decision to publish, or preparation of the manuscript.

### Grant Disclosures

The following grant information was disclosed by the authors:
Youth Innovation Promotion Association of Chinese Academy of Sciences: 2016305.
Major Project for New Agricultural Varieties Breeding of Zhejiang Province: 2012C12907-2.

### Competing Interests

The authors declare there are no competing interests.

## Author Contributions

- Junjun Yan performed the experiments, analyzed the data, prepared figures and/or tables, authored or reviewed drafts of the paper.
- Qiang Gao performed the experiments.
- Zongbin Cui and Guoliang Yang conceived and designed the experiments, approved the final draft.
- Yong Long conceived and designed the experiments, performed the experiments, analyzed the data, contributed reagents/materials/analysis tools, prepared figures and/or tables, authored or reviewed drafts of the paper, approved the final draft.

## Animal Ethics

The following information was supplied relating to ethical approvals (i.e., approving body and any reference numbers):

The animal protocol of this study was approved by the Institutional Animal Care and Use Committee of Institute of Hydrobiology (Approval ID: Y341131501).

## DNA Deposition

The following information was supplied regarding the deposition of DNA sequences:

The obtained upstream sequence counted from the first nucleotide of the initiation codon is 2,620 bp in length, is available at GenBank, accession number KY038927.

## Data Availability

Long, Yong; Yan, Junjun (2018): Molecular characterization of the giant freshwater prawn (*Macrobrachium rosenbergii*) beta-actin gene promoter. figshare. Fileset. https://doi.org/10.6084/m9.figshare.7076543.v3.

## Supplemental Information

Supplemental information for this article can be found online at http://dx.doi.org/10.7717/peerj.5701#supplemental-information.

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
