# Peer review of "Molecular characterization of the giant freshwater prawn (Macrobrachium rosenbergii) beta-actin gene promoter"

_PeerJ, doi:10.7717/peerj.5701_

## Round 0.1 · original submission · Minor Revisions

Please address all critical points raised by both reviewers and amend your manuscript accordingly.

Reviewer 1 ·

Basic reporting

I would like to commend the authors on presenting the data and results in a clear and unambiguous way.

I would like to see some improvements made to the introduction. In its present form, the introduction seems like a compilation of semi-disconnected paragraphs. By the end of the introduction, a connection can be seen. But I would suggest modifying the introduction to present a more connected background, which clearly justifies the relevance of the study, as well. Furthermore, the introduction goes into a lot of details about aspects such as humoral factors etc which are not appearing in the study at all. While it is always nice to have sufficient background and teach the reader more about the field, it could also help to shorten it and utilize that section to clearly mention the goals and gains of this study.

Some minor errors:

1. In the entire manuscript, please replace the apostrophe with the prime symbol for 5' etc.
2. Corrections needed:

Line 45: replace culture with aquaculture
line 57: "neutralize against invasion" -- please fix the sentence
Line 112: replace nest with nested
Line 127-28: change to: ..RNA sample was first treated with CIAP (...) to remove...
Line 246: replace islands with island
Line 249: Mba54 is used without introducing the concept. Please rearrange this section.
Line 345: rosenbergii was (space between the two words is missing)

Experimental design

The authors have presented a systematic study and have well characterized the promoter region. The molecular dissection, luciferase studies and identification of the various genetic elements has been very well conducted and presented.

Validity of the findings

I would again like to complement the authors for presenting a very neat study. They have thoroughly investigated and characterized the beta actin promoter.

I have one general comment. While reading the introduction, I got the impression that the use of Mbap over other promoters including Cbap would prove beneficial in driving gene expression. But, their activities are about the same. While it is useful that Mbap works as well as Cbap, I would recommend modifying the introduction accordingly.

Reviewer 2 ·

Basic reporting

no comment

Experimental design

no comment

Validity of the findings

no comment

Additional comments

In this manuscript, the authors study and characterize the gene promoter of actb1 in freshwater prawn, which can be a potential approach to protect prawn from disease and therefore benefit aquaculture economics. The author does sufficient and solid work as well as good interpretation of the experiment data. Meanwhile, the conclusions are clearly stated and reasonably drawn according to the experiment data. Overall, the paper is well-prepared without big issue of experiment design or data analysis. The only question is that the last part of study (comparison between prawn and carp actb1 promoter) doesn't fit the whole story of paper well due to the weak connection with previous parts. A better transition is necessary for its addition in the main context of paper. Otherwise, it should be deleted or moved into the supplementary part.
As far as the authors fix this, I would suggest acceptance for publication.

---

## Round 0.2 · accepted · Accept

Thank you for addressing all critiques raised by both reviewers and for revising the manuscript.

#